# Current forest carbon fixation fuels stream CO$_2$ emissions

A. Campeau [1], K. Bishop [2], N. Amvrosiadi [1], M.F. Billett[3], M.H. Garnett [4], H. Laudon [5], M.G. Öquist[5] & M.B. Wallin [1]

Stream CO$_2$ emissions contribute significantly to atmospheric climate forcing. While there are strong indications that groundwater inputs sustain these emissions, the specific bio-geochemical pathways and timescales involved in this lateral CO$_2$ export are still obscure. Here, via an extensive radiocarbon ($^{14}$C) characterisation of CO$_2$ and DOC in stream water and its groundwater sources in an old-growth boreal forest, we demonstrate that the $^{14}$C-CO$_2$ is consistently in tune with the current atmospheric $^{14}$C-CO$_2$ level and shows little association with the $^{14}$C-DOC in the same waters. Our findings thus indicate that stream CO$_2$ emissions act as a shortcut that returns CO$_2$ recently fixed by the forest vegetation to the atmosphere. Our results expose a positive feedback mechanism within the C budget of forested catchments, where stream CO$_2$ emissions will be highly sensitive to changes in forest C allocation patterns associated with climate and land-use changes.

[1] Department of Earth Sciences: Air, Water and Landscape, Uppsala University, Villavägen 16, 752 36 Uppsala, Sweden. [2] Department of Aquatic Sciences and Assessment, Swedish University of Agricultural Sciences, Lennart Hjelms väg 9, Uppsala 756 51, Sweden. [3] Biological and Environmental Sciences, Faculty of Natural Sciences, University of Stirling, Stirling FK9 4LA Scotland, UK. [4] NERC Radiocarbon Facility, Scottish Enterprise Technology Par, Rankine Avenue, East Kilbride, Glasgow G75 0QF Scotland, UK. [5] Department of Forest Ecology and Management, Swedish University of Agricultural Sciences, Skogsmarksgränd 17, 901 83 Umeå, Sweden. Correspondence and requests for materials should be addressed to A.C. (email: audrey.campeau@geo.uu.se)

The flushing of terrestrially-derived C through runoff can represent up to 30% of the annual C balance of forested ecosystems, of which greenhouse gases such as $CO_2$ hold a major contribution[1–3]. Once released from soils to surface waters, this $CO_2$ is rapidly evaded to the atmosphere as a result of physical gas exchange[4]. This lateral $CO_2$ flux is particularly relevant to headwater streams, which account for the bulk of surface water $CO_2$ emissions[5,6]. Streams thus contribute actively to atmospheric climate forcing by returning terrestrially sequestered $CO_2$ to the atmosphere. To date, the evidence supporting the terrestrial origin of stream $CO_2$ has relied solely on mass balance exercises; demonstrating that the pool of groundwater $CO_2$ is often sufficiently large to sustain stream $CO_2$ fluxes[7–9]. Explicit demonstrations of this link are still absent. Most of all, the specific biogeochemical pathways giving rise to stream $CO_2$, along with their associated timescales, have yet to be resolved. Without a clear assessment of the sources of lateral $CO_2$ fluxes, the terrestrial and aquatic components of catchment C budgets cannot be reconciled.

Soil and groundwater $CO_2$ can arise from several different biological sources, each confined across a spectrum between two main timescales[10,11]. Operating in the short timescales, is the current forest fixation of atmospheric $CO_2$, which fuels autotrophic root respiration[12] and heterotrophic mineralisation of recent photosynthates, transported to soils via throughfall, stemflow[13] and root leachates[14]. Alternatively, over longer timescales, saprotrophic decomposition embodies all forms of heterotrophic decomposition of older plant detritus and soil organic matter[15]. In groundwater, saprotrophic decomposition is supported mainly by dissolved organic C (DOC), which incorporates an assemblage of chemical properties and ages from the vegetation and soils traversed by the groundwater during its journey through the catchment[16,17]. Boreal forest catchments often comprise a peat-rich riparian zone, which serves as a repository of ancient soil organic matter that can support saprotrophic metabolism and may be remobilized through decomposition and runoff[18–20]. The source determination of lateral C fluxes may enable assessment of the vulnerability of these ancient C stocks[19,21,22]. Boreal forests also drive a considerable share of the global continental $CO_2$ sink[23], a process that is considered sensitive to a variety of anticipated disturbances[24,25]. The separation of timescales in the biological pathways governing lateral $CO_2$ fluxes in boreal forested catchments is thus critical information, since different $CO_2$ sources will likely follow distinct trajectories in response to environmental changes.

Here, we identify and apportion the sources of stream $CO_2$ with particular emphasis on the separation of timescales and biogeochemical pathways involved in the lateral $CO_2$ fluxes from an old-growth boreal forest catchment. We characterize the terrestrial and aquatic $CO_2$ sources via repeated measurements of the groundwater and stream water radiocarbon ($^{14}C$) content of $CO_2$ and DOC ($^{14}C$-$CO_2$, $^{14}C$-DOC). Our sampling was designed following a three-level Upslope-Riparian-Stream transect, repeated over three different occasions during the growing season, thus allowing to associate spatio-temporal $^{14}C$ patterns to different $CO_2$ sources. The transect sampling was complemented with a year-round characterisation of the stream water $^{14}C$-$CO_2$ and $^{14}C$-DOC, to further explore potential shifts in $CO_2$ sources over time. Automated sensors recording hourly $CO_2$ concentrations at each location along the transect and further downstream, allowed us to derive a complete annual C budget for this forested catchment using the age component of the lateral C fluxes to reveal links between the terrestrial and aquatic components. This study reveals that soil respiration, derived from the current forest C fixation, is the main source of stream $CO_2$ fluxes.

## Results

**Interpretation of $^{14}C$-contents.** Radiocarbon analysis can be used to determine the average age of $CO_2$ and DOC based on conventional $^{14}C$ dating techniques and represents one of the most robust approaches for the separation of respiratory processes in soils[10,11,26]. However, the $^{14}C$ content of gases or solute samples potentially originates from multiple combinations of sources, each with a different $^{14}C$-age, thus complicating the interpretation of the single average $^{14}C$-content. While the incorporation of post-bomb $^{14}C$ in the C cycle, resulting from the atmospheric testing in the 1950–60s, precludes a linear interpretation of $^{14}C$-content, it can provide clear evidence of carbon fixed from the atmosphere post ~AD1955 (i.e., when $^{14}C$ concentration > 100%). Here, we specifically avoided referring to the measured $^{14}C$ contents in terms of age from conventional $^{14}C$ dating and instead focused our analysis on the relative differences in $^{14}C$ contents between C species, in our case $CO_2$ and DOC, as well as their changes over time and space to help define the stream C sources in connection to terrestrial processes.

**Stream water $^{14}C$-$CO_2$ and $^{14}C$-DOC.** Stream water $^{14}C$-$CO_2$ was surprisingly constant throughout the year, ranging from 102.5 to 105.3 %modern ($n = 11$) (Fig. 1a, Supplementary Table 1). In comparison, the stream water $^{14}C$-DOC was more variable and more $^{14}C$-enriched, ranging from 103.5 to 112.2 %modern ($n = 8$) (Fig. 1a, Supplementary Table 2). Stream water $^{14}C$-DOC was negatively related to the riparian water table position, indicating that more superficial water tables corresponded to more $^{14}C$-enriched DOC in the stream waters (Fig. 1b). There was no significant relationship between stream water $^{14}C$-$CO_2$ and any of the measured variables included in the study, for example: temperature, discharge, water table position, C concentrations, net ecosystem exchange (NEE) or photosynthetic photon flux density (PPFD) (all $p > 0.05$, Fig. 1).

**Groundwater $^{14}C$-$CO_2$ and $^{14}C$-DOC.** The patterns in $^{14}C$-$CO_2$ and $^{14}C$-DOC in the connecting groundwater were generally similar to those of the stream waters. As such, groundwater $^{14}C$-$CO_2$ was also remarkably homogenous across locations, depths and sampling dates. All samples were enriched in post-bomb $^{14}C$, together ranging from 101.1 to 106.6 %modern ($n = 7$), with the exception of one sample collected in the riparian deep soil water in August, where $^{14}C$-$CO_2$ was 99.0 %modern (Fig. 2, Supplementary Table 2). Differences in $^{14}C$-$CO_2$ between upslope and riparian groundwater were not significant ($p = 0.6$), despite a near doubling of the $CO_2$ concentrations between the two locations. In contrast, the range in groundwater $^{14}C$-DOC was much larger than for $^{14}C$-$CO_2$. Most of the groundwater $^{14}C$-DOC were enriched in post-bomb $^{14}C$, ranging from 100.8 to 116.8 %modern ($n = 9$), but there were two groundwater samples for which $^{14}C$-DOC was remarkably depleted (49.7 and 67.8 %modern, respectively) (Fig. 2, Supplementary Table 2). Both were collected in the upslope deep location, in August and October. Excluding these two groundwater samples, the $^{14}C$-DOC was significantly negatively correlated with the DOC concentration across the groundwater and stream waters (Fig. 3). There was no similar relationship between $^{14}C$-$CO_2$ and $CO_2$ concentrations in either the groundwater or stream water (Fig. 3).

The $^{14}C$-$CO_2$ and $^{14}C$-DOC showed little correspondence in both groundwater and stream waters, with the DOC being on average 6 %modern more $^{14}C$-enriched than the $CO_2$ across all discrete samples ($n = 9$), with the exception of the upslope deep groundwater in August and October (Figs. 1a, 2). In these groundwater, the difference in $^{14}C$ content between $CO_2$ and DOC was even larger, corresponding to 55 and 33 %modern for

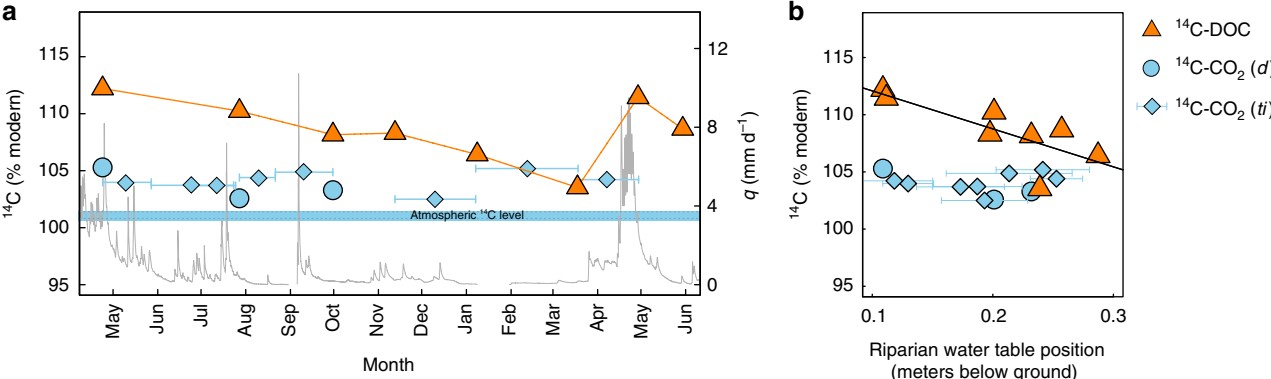

**Fig. 1** Hydrograph overlayed with stream water $^{14}$C-$CO_2$ and $^{14}$C-DOC time series. **a** Hydrograph showing the stream specific discharge (q) in mm d$^{-1}$ (grey line) and stream water $^{14}$C-$CO_2$ (blue circles (discrete samples (d)) and bars with diamonds (time-integrated samples (ti))) and $^{14}$C-DOC (orange triangles), all expressed in %modern for the period May 2015–June 2016. The blue rectangle indicates the range in atmospheric $^{14}$C-$CO_2$ content in the northern hemisphere during 2015–2016, according to Graven et al. [68] **b** scatterplot showing the stream water $^{14}$C-$CO_2$ and $^{14}$C-DOC as a function of riparian water table position (in metres below ground surface). The solid line represents the least square linear regression model, $^{14}$C-DOC = WT$_R$ × −33.3 + 115.4, $R^2$ = 0.53, $p$ = 0.02. The original $^{14}$C data are listed in the Supplementary Table 1

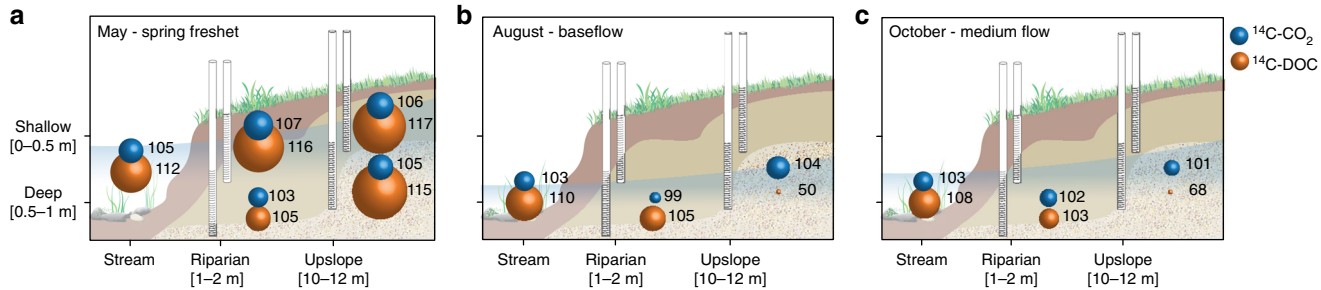

**Fig. 2** Groundwater and stream water $^{14}$C-$CO_2$ and $^{14}$C-DOC at the transect scale. Catchment schematic (unscaled) illustrating the $^{14}$C-$CO_2$ (blue spheres) and $^{14}$C -DOC (orange spheres) expressed in %modern. The size of the spheres is proportional to the $^{14}$C content and specific values are written on the right side of each sphere. The measurements were made along the Upslope-Riparian-Stream transect, at shallow (0–0.5 m) and deep (0.5–1 m) groundwater depths, in **a** May 2015, **b** August 2015, **c** October 2015. The original $^{14}$C data are listed in the Supplementary Table 2

August and October respectively, where DOC was suddenly much older than $CO_2$ (Fig. 2). In the riparian deep groundwater, the $^{14}$C-$CO_2$ and $^{14}$C-DOC matched most closely and also showed the least temporal variability (only ~2 %modern across the three-sampling occasions despite a near doubling of the DOC concentrations between sampling occasions) (Fig. 2). In the stream waters, the differences in $^{14}$C content between $CO_2$ and DOC, was largest during the spring freshet, but decreased during winter base flow conditions (Fig. 1a).

**$CO_2$ and DOC concentrations**. The average $CO_2$ concentrations doubled between the upslope mineral soils and the riparian organic-rich soils ($d$ = 0.45) (Fig. 4e, f). In the riparian soils, the $CO_2$ concentrations were similar between the shallow and deep layers (17.7 ± 3.5 and 14.9 ± 3.3 mg C L$^{-1}$, respectively $n$ = 8028 ($d$ = 0.2)), while in the upslope soils, the $CO_2$ concentrations were significantly lower in the shallow compared with the deep layers (4.6 ± 2.0 mg C L$^{-1}$ $n$ = 707, and 11.7 ± 3.3 mg C L$^{-1}$ $n$ = 8027 ($d$ = 1.2), respectively) (Fig. 4e, f). Stream water $CO_2$ concentrations during the open water season were significantly higher in the location adjacent to the transect (3.4 ± 1.1 $n$ = 3496), compared with the stream gauging station, located 250 m downstream (2.8 ± 0.9 $n$ = 3235), ($d$ = 1.3) (Fig. 4d). Year-round hourly stream water $CO_2$ concentrations were recorded at the downstream location, but showed no significant difference between the ice-covered and open-water period ($d$ = 0.48, annual mean 2.8 ± 1.4 $n$ = 8311) (Fig. 4d). The stream water $CO_2$

concentrations never exceeded the groundwater $CO_2$ concentrations in the riparian or upslope location (Fig. 4d–f).

Groundwater DOC concentrations were on average five times higher in the riparian compared with the upslope groundwater (39.2 ± 10.4 mg C L$^{-1}$ $n$ = 11, and 7.4 ± 6.7 mg C L$^{-1}$ $n$ = 8, respectively, $p$ < 0.0001) (Fig. 4e, f). DOC concentrations were similar across the two groundwater depths, in both the riparian and upslope soils ($p$ = 0.06, $p$ = 0.5, respectively). The riparian DOC concentrations increased steadily between May and October, by 87 and 33% in the shallow and deep layers, respectively (Fig. 4e). This increase was not as clear in the upslope soils. The stream water DOC concentrations (19.8 ± 4.8 mg C L$^{-1}$ $n$ = 8) were significantly lower and never exceeded the riparian groundwater DOC concentrations ($p$ ≤ 0.0001) (Fig. 4d, e). Together, the trends in DOC and $CO_2$ concentrations along the upslope-riparian-stream transect resulted in a progressive shift, from a slight dominance of $CO_2$ over DOC in the upslope groundwater (average $CO_2$-C:DOC = 1.5), to a clear dominance of DOC over $CO_2$ in the riparian groundwater (average $CO_2$-C:DOC = 0.4), which was more pronounced in the stream waters (average $CO_2$-C:DOC = 0.1).

**Hydro-climatic conditions and catchment C budget**. The NEE (determined by Eddy Covariance) of the forest ecosystem in the catchment for the study year was −205 g m$^{-2}$ yr$^{-1}$ (Fig. 5). The annual runoff for the catchment during the study year was 257 mm, representing about half of the annual precipitation

(507 mm). The stream specific discharge ($q$) ranged from 0.036 to 10.8 mm d$^{-1}$ (Fig. 1a). The modelled upslope and riparian water export within the upper one metre was 236 and 243 mm, respectively, which is similar to the catchment annual runoff (Supplementary Fig. 1c). Runoff through the shallow soil depths (0–0.5 m) contributed to 58 and 73% of the total runoff in the upslope and riparian location, respectively (Supplementary Fig. 1c).

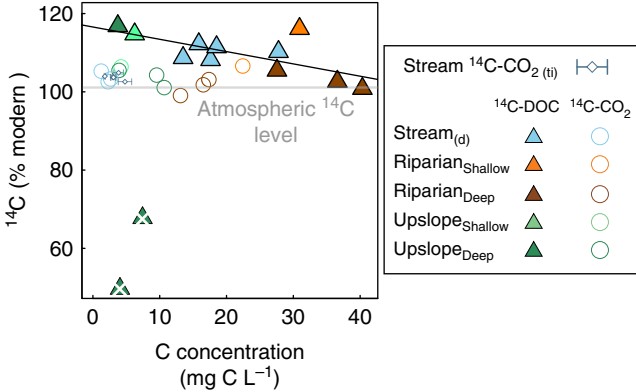

**Fig. 3** Mass controls on $^{14}$C-$CO_2$ and $^{14}$C-DOC in groundwater and stream water. Scatterplot showing the groundwater and stream water $^{14}$C-$CO_2$ and $^{14}$C-DOC in %modern as a function of their respective concentrations in mg C L$^{-1}$. The solid line represents the least square linear regression model $^{14}$C-DOC = DOC × −0.31 + 116.6, $R^2$ = 0.45, $p$ = 0.01, excluding the two severely depleted $^{14}$C-DOC values from the upslope deep location in August and October identified with the white cross. The grey line indicates the range in atmospheric $^{14}$C-$CO_2$ content in the northern hemisphere during 2015–2016, according to Graven et al. [68]

The annual $CO_2$ export from the riparian location was more than double that of the upslope location (Fig. 5). The contribution of the riparian soils was even more substantial for the annual DOC export, which was more than five times larger compared with the upslope site (Fig. 5). Downstream $CO_2$ export represented only 13% of the initial riparian groundwater $CO_2$ export, which can be attributed to rapid $CO_2$ evasion to the atmosphere. The downstream DOC export was more comparable to the riparian groundwater DOC export, representing about 70% of the initial flux. The cumulative age of the lateral C fluxes was 104 and 110 %modern for $CO_2$ and DOC, respectively. The difference in $^{14}$C content between the annual $CO_2$ and DOC exported from the catchment could be explained by a 75% contribution from currently fixed $CO_2$ from the atmosphere (i.e., during the last growing season (2015–2016), with the remaining fraction originating from the bulk DOC mineralization.

## Discussion

This study provides, to our knowledge, the first explicit evidence that stream $CO_2$ fluxes are sustained by currently fixed $CO_2$ from atmosphere via the forest vegetation's photosynthetic activity (i.e., during the last growing season (2015–2016). The first piece of evidence supporting our interpretation was the persistent gaps between $^{14}$C-$CO_2$ and $^{14}$C-DOC in groundwater and stream water, highlighting a major disconnect in the cycling of the two C species. Secondly, the homogeneity of the $^{14}$C-$CO_2$ in groundwater and stream water, which remained systematically close to the current atmospheric $^{14}$C level, indicated that $CO_2$ was sustained by a large and steady source, likely associated with current photosynthesis. Previous studies have provided indications that groundwater inflow of soil-derived $CO_2$ is sufficient to support stream $CO_2$ sources[7–9]. However, explicit demonstrations of the

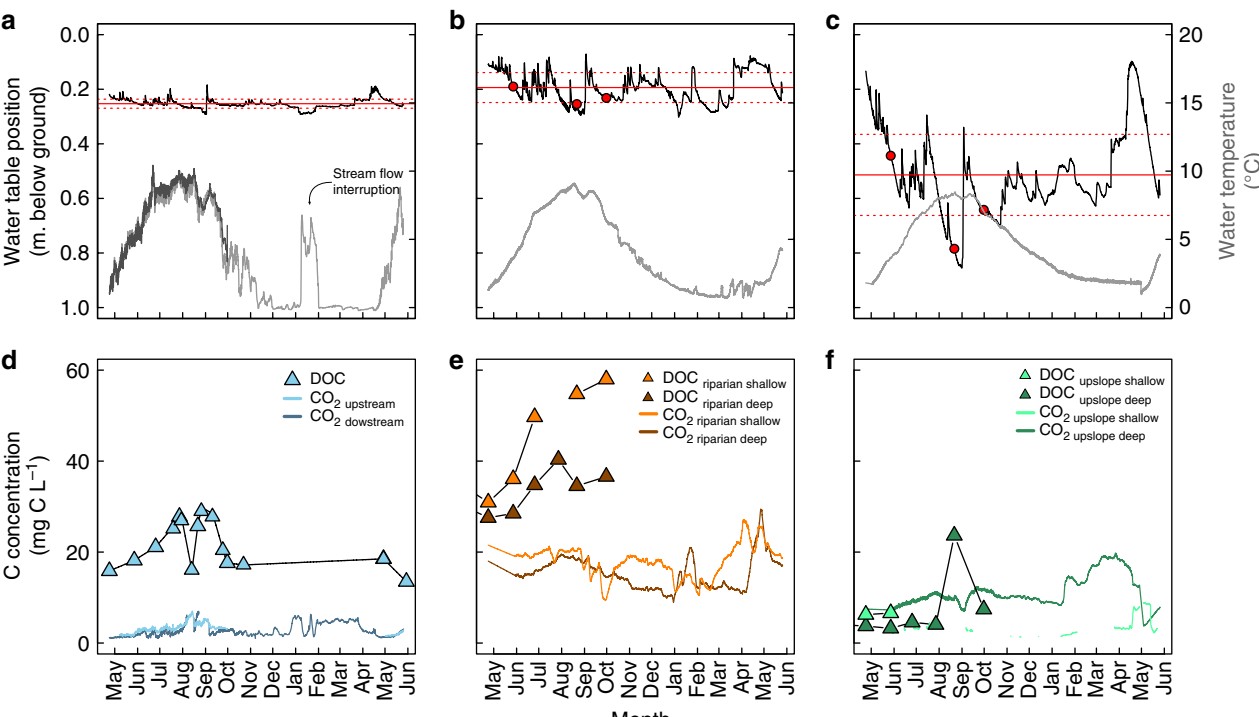

**Fig. 4** Time series of hydro-climatic conditions and C concentrations across the transect. **a–c** Time series of water table position (black line) with its annual mean and standard deviation (full and dotted red lines, respectively) and the water temperature (grey line) in **a** the stream water, **b** riparian groundwater, and **c** upslope groundwater. The groundwater $^{14}$C sampling occasions are identified by red circles. **d–f** Time series of $CO_2$ concentration and DOC concentration in the **d** stream water, **e** riparian groundwater, and **f** upslope groundwater. Time series are presented for the period from May 2015–June 2016

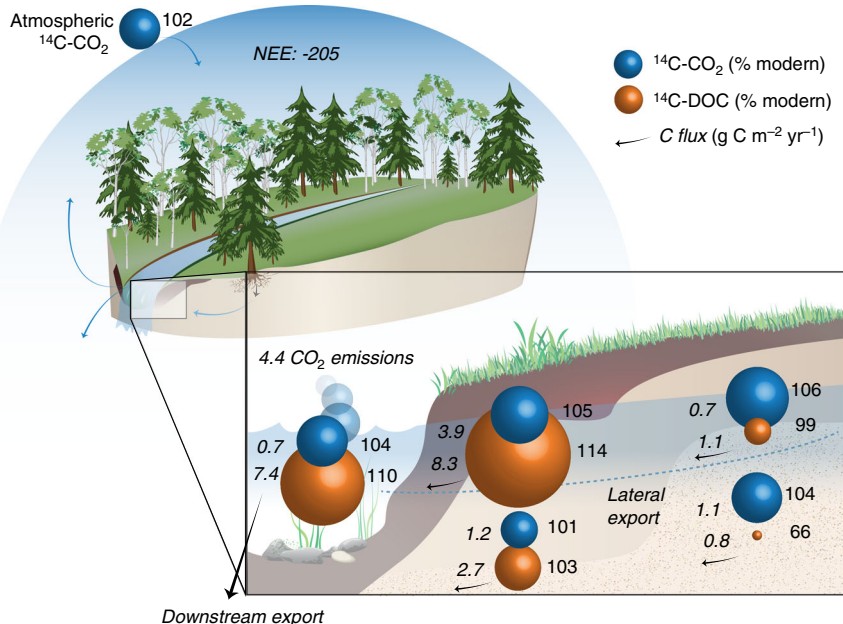

**Fig. 5** Schematic of the annual catchment C budget for the studied year. This schematic (unscaled) includes the forest NEE and the lateral and downstream $CO_2$ and DOC export and stream $CO_2$ emissions, along with their cumulative annual [14]C contents represented by the sizes of the spheres (blue ([14]C-$CO_2$) and orange ([14]C-DOC)), at the three locations along the Upslope-Riparian-Stream transect, and depth (shallow [0–0.5 m] and deep [0.5–1 m]). Annual C export rates (g C m$^{-2}$ yr$^{-1}$) are written in italic on the left side of each sphere, while the [14]C contents are written on the right side

link between soil and stream $CO_2$, along with determination of the biogeochemical pathways involved, were still lacking. Our results imply that stream $CO_2$ fluxes are cycled rapidly, and likely provide a fast pathway for returning $CO_2$ fixed from the atmosphere by the forest vegetation during this year's growing season. The main implication of this work is that anticipated alterations in boreal forest growth and ecosystem level C allocation patterns, driven by climate change and other disturbances, will produce a rapid response in the stream $CO_2$ fluxes, since both processes are tightly linked by the current forest activity.

The systematic offset between the [14]C-DOC and [14]C-$CO_2$ in groundwater and stream water is key evidence of the limited overlap in their respective sources and controls (Figs. 1a, 2). The substantial variability in groundwater and stream water [14]C-DOC, which contained both severely [14]C-depleted and [14]C-enriched post-bomb values (Figs. 1, 2), indicated that DOC arises from more diverse sources than $CO_2$ and is cycled more slowly, up to millennia. There was a clear connection between the [14]C-DOC and hydrological retention and flowpaths (Figs. 1b, 3), as suggested by previous studies in this catchment using independent methods[19,27,28]. Activation of fast flowing superficial flowpaths[29] was associated with the transport of modern [14]C-DOC, with rising water tables leading to an increase proportion of post-bomb [14]C and dilution of the DOC, for example during spring freshet (Figs. 1b, 3). These dynamic superficial flowpaths are supplemented by intermittent activations of deeper flowpaths associated with longer water retention times[30] and the transport of aged-DOC, for example, the upslope groundwater later in the growing season (Figs. 2, 3). Despite these profound changes in hydrological flowpaths across locations and seasons, the groundwater and stream water [14]C-$CO_2$ remained relatively unchanged (Figs. 1, 2). In fact, the groundwater [14]C-$CO_2$ was similar between the upslope mineral soils and the riparian organic soils, despite major contrasts in soil properties and contributing flowpaths (e.g., a near doubling of the $CO_2$ concentrations and a five times increase in DOC concentrations (Fig. 4e, f), and a shift from severely [14]C-depleted to post-bomb enriched DOC between

the two locations later in the growing seasons (Fig. 3). This suggest that the sources governing groundwater $CO_2$ can override these dynamics in hydrological flowpaths and soil chemistry.

Inconsistencies between the [14]C content of DOC and $CO_2$ have been reported in other catchments including the Amazon river network[31] and various peatland dominated catchments[32–34] (Supplementary Fig. 2). Other studies comparing the [14]C content of DIC and DOC in surface waters also concur with these observations[35]. Although the form of these isotopic inconsistencies may vary across catchments, the [14]C-DOC often reveals greater levels of post-bomb C than the $CO_2$, suggesting more association with moderately old organic C reservoirs[31,35]. Severely [14]C-depleted $CO_2$ and DIC in surface waters are also more frequently reported than DOC, but these are typically connected to weathering of carbonate-containing minerals[36,37]. Such geological sources of $CO_2$ are absent in this catchment, as indicated by the $\delta^{13}$C-$CO_2$ values that were consistent with the C3 plant metabolic pathway (Supplementary Fig. 2[38]). A considerable number of observations in the literature demonstrate a close agreement between the surface water [14]C-$CO_2$ and the current atmospheric [14]C level, together with a clear photosynthetic $\delta^{13}$C value[31,35]. These observations, comply with ours, and suggests that surface water $CO_2$ sources may often arise from rapid C cycling processes within catchment soils.

The uniformity of [14]C-$CO_2$ and its similarity with the current atmospheric [14]C level, underlined that groundwater and stream water $CO_2$ was sustained by a large and steady source, with rapid turnover times and omnipresent across this catchment (Figs. 1a, 2). Forest C fixation can fuel groundwater $CO_2$ via autotrophic root respiration[12] or the transport of recent photosynthates to soils by root exudates[14], throughfall and stemflow[13], thereafter mineralized by the soil microbial communities. Invasion of atmospheric $CO_2$ in groundwater and stream water could not explain this close agreement between the [14]C-$CO_2$ content of the current atmospheric [14]C level, as indicated by high $CO_2$ concentrations in soil and stream waters, consistently above atmospheric saturations (Fig. 4d–f) and the low $\delta^{13}$C-$CO_2$ values,

conformed with C3 derived organic matter (Supplementary Fig. 2[38]). The stability of the $^{14}C$-$CO_2$ content was remarkable, considering the dramatic seasonal changes in groundwater and stream water C concentrations, forest productivity, precipitation, runoff and temperature over the year. In fact, the $^{14}C$-$CO_2$ was independent of all measured environmental variables monitored in this study and showed no connection to seasonal patterns, further supporting a steady source (Figs. 1–3). Seasonal changes in hydro-climatic conditions are major controls on forest C fixation, soil respiratory processes[39,40] and hydrological connectivity between soils and streams[41], but the large amount of $CO_2$ fixed by the forest vegetation during the growing season is likely sufficient to support the lateral $CO_2$ export throughout the year (Fig. 5).

Based on the cumulative $^{14}C$ content of stream $CO_2$, an estimated 75% of the lateral $CO_2$ export from the catchment possibly originated from the forest C fixation during the last growing season, with the remaining fraction arising from saprotrophic respiration, subsidized by the cumulative bulk $^{14}C$-DOC export (Fig. 5). This rough estimate agrees well with studies partitioning the vertical soil $CO_2$ fluxes, where autotrophic root respiration contributes 50–64% of the total soil respiration in various forested catchments[12,15], as well as in a nearby boreal forest catchment[26]. This flux is likely supplemented by heterotrophic mineralization of recent photosynthates, but the relative importance of these processes cannot be separated here. While there is general consensus that autotrophic root activity makes up a major fraction of vertical soil $CO_2$ efflux[42], there is also awareness that mineralization of root exudates and DOC transported via throughfall and stemflow may not be adequately quantifiable due to rapid turnover rates[14]. The specific contribution of these different biological pathways to lateral $CO_2$ export remains to be further investigated, but our results support a steady and widespread connection to rapidly cycling $CO_2$ sources fuelling lateral $CO_2$ export. Only a small fraction (ca. 2%) of the net $CO_2$ fixed from the atmosphere by the vegetation that year (NEE: $-205\,g\,C\,m^{-2}\,yr^{-1}$) was lost through the lateral $CO_2$ export (Fig. 5). This is consistent with mass balance estimates from the same catchment reported by Öquist, Bishop[1] during years with similar precipitation. Considering that precipitation and runoff were low during our study year, the fraction of currently fixed $CO_2$ lost through lateral $CO_2$ export in this forested catchment is likely higher in other years (up to 9%[1]).

Since the bulk DOC pool contains a mixture of C ages, the $^{14}C$-DOC in this catchment could reflect refractory DOC compounds, not mineralized during transit, masking an underlying connection between $^{14}C$-$CO_2$ and $^{14}C$-DOC. Further studies are needed to assess the age-composition of DOC, but this does not invalidate that the current atmospheric $^{14}C$-$CO_2$ was reflected in the stream and groundwater of this boreal forest catchement. Saprotrophic respiration may be limited in theses $CO_2$-rich groundwater, owing to lack of oxygen[38] and short water transit time in the transiently saturated zone[43,44]. The best agreement between $^{14}C$-$CO_2$ and $^{14}C$-DOC was observed in the deep riparian groundwater, which is found below the dominant source layer. The latter is responsible for the majority of DOC[19], DIC[41] and water export to the stream[43]. Longer residence time in the deeper groundwater may promote DOC mineralization, likely through fermentative processes[38], hence a closer overlap between $^{14}C$-$CO_2$ and $^{14}C$-DOC.

Closer agreements between the $^{14}C$-DIC and $^{14}C$-DOC than those observed in this study have sometimes been reported in larger river catchments and lakes[45–47]. This highlights that the terrestrial DOC export may remerge as an important source sustaining $CO_2$ emissions further downstream; when connectivity with the catchment soils decreases[8,48] and longer water residence time allows for in-situ mineralisation to occur[49]. Both aged and modern DOC were exported from this catchment during the study year, but our results do not indicate any significant incorporation of severely $^{14}C$-depleted or highly $^{14}C$-enriched post-bomb DOC in the lateral $CO_2$ export. Emerging new research is now demonstrating that aged-DOC may be bioavailable for freshwater microbial communities leading to the production of aged-$CO_2$[22,50,51]. The remobilization of ancient DOC in the upslope soils of this catchment is a concern, and more studies are needed to assess its sources and fate. While previous studies have shown that aquatic DOC mineralization remains low across the Krycklan catchment[52–54], there is also mounting evidence of deeper groundwater contribution increasing further downstream and potentially transporting aged-DOC[19,48,55]. Further studies are needed to address the generality of our findings, and the potential mineralization of aged-DOC within higher order rivers and streams.

This study reveals that stream $CO_2$ fluxes are fuelled by the current forest C fixation and its associated soil respiration processes in a boreal forest catchment. This close connection between the forest C sink and lateral $CO_2$ fluxes had already been put forward based on inter-annual coupling in flux measurements[1]. Our study provides further description of the mechanistic underpinning of this connection between terrestrial and aquatic C fluxes. The lateral $CO_2$ export rapidly mobilises a significant fraction of the C currently fixed from the atmosphere by the forest vegetation. Groundwater $CO_2$ inputs support surface waters $CO_2$ emissions across multiple types of aquatic ecosystems[7–9], thus suggesting that rapidly cycling $CO_2$ sources may have a widespread contribution to aquatic $CO_2$ emissions. Owing to the prevalence of root respiration and mineralization of recent photosynthates in forested soils, our results may also be generalized across a large number of forested ecosystems and biomes. Forest C balance and ecosystem level C allocation patterns in the northern high latitudes are vulnerable to a large number of disturbances including global warming, increased forest fire frequency, insect outbreaks and industrial and commercial exploitation[56–58]. Stream $CO_2$ emissions will quickly feedback on these disturbances, because of the speed of stream $CO_2$ cycling and its close connection to the current forest activity.

## Methods

**Catchment characteristics.** The study was conducted in a 0.13 km$^2$ catchment located in northern Sweden within the Krycklan Catchment Study (64°14′N, 19°46′E)[59]. The catchment has been heavily studied for more than two decades and is occasionally referred to as "Västrabäcken" or "C2" across the literature. The catchment is almost completely forested (99%), with Scots pine (*Pinus sylvestris*) (64%) and Norway spruce (*Picea abies*) (36%). The active root depth is mostly distributed above the average groundwater table position[60]. The average tree stand age is 103 years old[59]. Man-made ditching of the stream to improve forest productivity occurred about a century ago. The stream is adjacent a peat-rich riparian zone, with the soil profile consisting of ~70 cm thick peat transitioning to the underlying till at ~90 cm depth. The age of the accumulated solid peat ranges from modern near the surface to 2810 years BP at 70 cm depth (Bishop, unpublished data). The organic soil content is > 80% in the riparian zone, which is considerably higher than the upslope podzols (< 5%)[61]. The latter is composed of well-developed iron podzols on sandy till, comprising a 5 cm humus layer at the surface, overlying a 12 cm thick sandy bleached E-horizon and a 60 cm thick B-horizon[61]. The underlying bedrock is composed predominantly of base-poor Sveco-fennian metasediments-metagraywacke and holds no known carbonate containing minerals. Carbonate alkalinity is rather produced by weathering of silicate minerals[38].

The climate is cold temperate humid and bears a persistent snow cover from November to April. The 30 year mean annual precipitation is 640 mm (1981–2010), of which 35% falls as snow[59]. The annual precipitation during our study year (507 mm) corresponds to the bottom 6% of the previous 30 years observations (1981–2012). The 30 years mean annual, July and January temperatures are +1.8, +14.7 and −9.5 °C, respectively[59]. The annual peak stream discharge in the region typically occurs during spring in connection to snow melt, but storm events during summer and autumn can also generate peak flows in some years. The winter is typically dominated by low flow conditions.

**Soil and stream instrumentation**. Groundwater and stream water sampling was carried out following an upslope-riparian-stream transect, as described in ref. [38,41]. Groundwater wells were installed along the assumed hydrological flowpaths, with a first set located at 1–2 m (riparian) and the second at 10–12 m (upslope) distance from the stream. The 10-year mean water travel time for the entire catchment is estimated at 690, ranging between 470 and 2064 days[62]. The water turnover time, from the water divide to the stream and with depths ranging between 0.5 and 3 m, is estimated to 4.6 years[43]. The area represented by the groundwater transect was estimated at 2540 m$^2$ with an average width of 17.7 m, occupying 2% of the total catchment area[43]. The estimated time for water to exit this transect is in the scale of a month (near the upslope mineral soils) to hours (near the stream)[43].

The groundwater characterisation at both locations was focused to the upper one metre of the soil profile, where most of the runoff generation is confined[43]. The groundwater installation comprised a set of wells with screening of the upper 0–0.5 m (shallow) and lower 0.5–1 m (deep), respectively. For each of the two locations and depths, two identical sets of groundwater wells were installed, with one allowing manual sampling of the groundwater and the other containing sensors for continuous reading of groundwater temperature and $CO_2$ concentration. Alongside, a fifth well was installed at each of the upslope and riparian locations where water table position was recorded.

Continuous reading of dissolved $CO_2$ concentration at each location and depth was enabled using Vaisala CARBOCAP GMP221 nondispersive infrared (NDIR) $CO_2$ sensors (range 0–3%, 0–5%, in the stream and groundwater, respectively, except for the upslope shallow groundwater where the range was 0–1%). The sensors were enclosed inside a water-tight, gas-permeable Teflon membrane (PTFE) and sealed with Plasti Dip (Plasti Dip international, Baine, MN, USA) to ensure that the sensor was protected from water, but remained exposed to dissolved gas. The groundwater $CO_2$ concentration in the riparian and upslope location at the deep and shallow depth was recorded during the full year. The upslope shallow groundwater well was completely dry for 245 days out of the studied year, but all other sensors remained below the groundwater table at all times. The measurements in the stream adjacent to the groundwater wells were restricted to the open-water season. However, year-round measurements of stream water $CO_2$ concentration was undertaken in a heated dam house (C2), located 250 m downstream from the soil transect location. The continuous reading of groundwater and stream water $CO_2$ concentrations were validated against manual snapshot measurements, using an acidified headspace method[63], which showed an average 11 and 9% difference in the groundwater and stream water, respectively. All three stations along the transect (upslope, riparian and stream) were instrumented with pressure transducers (MJK 1400, 0–1 m, MJK Automation AB) recording water table height, and temperature sensors (TO3R, TOJO Skogsteknik). Groundwater temperature was measured at 0.8 m from the ground surface, and stream water temperature was recorded in two locations, adjacent to the transect and 250 m downstream. All continuously measured data were collected hourly and stored on external data loggers (CR1000, Campbell Scientific, USA).

Discharge was determined at a V-notch weir in the downstream heated dam house at C2[59]. Stream discharge was determined by applying stage height-discharge rating curves to hourly water level measurements. The discharge for each of the four groundwater sampling points (riparian/upslope, deep/shallow) was modelled according to Amvrosiadi, Seibert[43]. To calculate total discharge through the riparian and upslope profiles, Darcy's law was applied in combination with the transmissivity profiles, and the local water table gradients[43]. The fraction of discharge flowing through the shallow (0–0.5 m) and deep (0.5–1 m) soil layers was estimated based on the same transmissivity profiles.

**Soil and stream water chemical analysis**. Groundwater and stream water samples were collected monthly for DOC at each location and depth during the ice-free season (May–November 2015). The DOC concentration was analysed from 10 ml of ground and stream water, filtered through glass-fiber Whatman GF/F filters (0.7 μm) in the field and stored in high-density polyethylene bottles. Prior to analysis, samples were acidified and sparged to remove inorganic carbon. The samples were analysed using a Shimadzu Total Organic Carbon Analyzer TOC-V$_{CPH}$, following storage at 4 °C for 2–3 days[64].

**Radiocarbon analysis**. The radiocarbon sampling was carried out in two phases, first a repeated catchment scale characterisation of groundwater and stream water in May (spring), August (summer) and October (autumn), and consecutively, a complete year characterisation in the stream waters. In total, 21 samples were collected and analysed for $^{14}C$-$CO_2$ and 16 for $^{14}C$-DOC in groundwater and stream water between May 2015 and June 2016. The stream water samples were collected directly adjacent to the groundwater transect during the open water periods and downstream in the heated dam house (C2) during the ice-covered periods for accessibility. Simultaneous measurement of $^{14}C$-$CO_2$ at both stations in July 2015 showed close correspondence, with only a <0.1 %modern difference, which was within the range of measurement precision. The hillslope transect sampling included all four groundwater locations and depths; riparian/upslope and deep/shallow. However, the available water volume in the shallow

riparian and upslope locations was insufficient for radiocarbon analysis in August and October.

Sample collection for $^{14}C$-$CO_2$ included two different methods, one using manual spot measurements applied to groundwater and stream water ($n = 11$) and the other allowing for time-integrated sampling in the stream water ($n = 10$). Manual spot measurements of $^{14}C$-$CO_2$ were carried out with the super headspace method whereby manually equilibrated $CO_2$ samples were trapped onto molecular sieve cartridges (MSCs) (see ref. [65] for further details). The integrated measurements of $^{14}C$-$CO_2$ were performed using passive samplers comprising MSCs installed below the water surface, which slowly collects stream water $CO_2$ over extended time periods (see ref. [66] for further details). The samplers were based on the MSC described above, but attached to a gas permeable hydrophobic filter (Accurel PP V8/2 HF tubing; Membrana GmbH, Germany[67]). These passive samplers were deployed for periods ranging from 26 to 72 days; collectively they cover more than a full year (May 2015–June 2016). The trapping capacity of the MSC was never exceeded (< 100 ml $CO_2$). Unfortunately, two time integrated $^{14}C$-$CO_2$ samples were discarded due to contamination that resulted from cracks in the MSC glass casing during deployment or transportation that caused stream water or air contamination of the sample. At the NERC Radiocarbon Facility (East Kilbride, UK), $CO_2$ samples were recovered from the MSCs by heating and cryogenically purified.

The $^{14}C$-DOC analysis was performed on 1 L samples of groundwater and stream water collected in acid-washed glass bottles. The samples were filtered in the laboratory through 0.7 μm glass fibre filters, rotary evaporated and freeze-dried. Acid-fumigation of samples was undertaken to guard against carbonate contamination, and the dried DOC was combusted to $CO_2$ in an elemental analyser (Costech ECS 4010, Italy) and cryogenically recovered. Manual spot measurements of stream water $^{14}C$-DOC were taken at each change of the passive $^{14}C$-$CO_2$ samplers in order to characterize cumulative $^{14}C$-DOC under a large range of hydrological conditions.

All radiocarbon samples were converted to graphite using Fe-Zn reduction and measured by accelerator mass spectrometry at the Scottish Universities Environmental Research Centre (East Kilbride, UK). Stable carbon isotope measurement ($\delta^{13}C$) was performed on an aliquot of the recovered $CO_2$ using isotope ratio mass spectrometry (IRMS; Thermo-Fisher Delta V, Germany) and reported relative to the Vienna PDB standard. All radiocarbon results were normalised to a $\delta^{13}C$ of −25 ‰ using the measured $\delta^{13}C$ values, and expressed as %modern and conventional radiocarbon age (years before present (BP), where 0 BP = AD 1950), with ± 1σ analytical precision. The passively collected $^{14}C$-$CO_2$ samples were additionally corrected for the +4.2 ‰ isotopic fractionation effect caused by the gas trapping into the molecular sieves[66]. The northern hemisphere atmospheric $^{14}C$-$CO_2$ content during the study period ranged from 101.4 to 100.8 according to ref. [68].

**Catchment C budget and statistical analysis**. The contribution of currently fixed $CO_2$ from the atmosphere by the forest vegetation during the growing season of the studied year (2015–2016; Atm$^{14}C$ (%)) was estimated using a simple two-endmember mixing model, solving for the mass of current atmospheric $^{14}C$-$CO_2$ (Atm$^{14}C$) required to explain the observed gap between the cumulative lateral $^{14}C$-$CO_2$ and $^{14}C$-DOC export as follows:

$$\text{Atm }^{14}C\,(\%) = \left( ^{14}C - CO_2 - ^{14}C - DOC \right) / \left( Atm^{14}C - DOC \right) \times 100$$

The forest NEE and PPFD were obtained from the ICOS data from the Svartberget site (http://www.icos-sweden.se/). The groundwater C export for each soil depth and location was estimated by combining the modelled specific discharge according to Amvrosiadi, Seibert [43], along with the measured hourly $CO_2$ concentrations and the interpolated monthly point measurements of DOC concentrations. The total $^{14}C$ content export at the upslope, riparian and stream locations was estimated by calculating the weighted average $^{14}C$ content of both $CO_2$ and DOC export over the full year. Significant differences in soil or stream water chemistry were tested using the non-parametric Wilcoxon test, with p-values reported in brackets in the result sections. For large sample sizes, such as hourly measurements of $CO_2$ concentrations, statistical differences between means were tested using Cohen's d test for effect size. Differences were considered statistically significant when $p$-value < 0.01. Mean values followed with the standard deviation and the number of observations are presented in brackets in the text. All analyses were performed using R Core Team (2013). R: A language and environment for statistical computing. R Foundation for Statistical Computing, Vienna, Austria. URL http://www.R-project.org/.

## Data availability

All radiocarbon data are presented in Supplementary Tables 1 and 2. The accompanying datasets and codes generated analysed during the current study are available from the corresponding author on reasonable request.

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

## Acknowledgements

This study was supported by the Swedish Research Council (contract: 2012-3919 to K. Bishop), the Natural Environment Research Council (NERC) Radiocarbon Facility NRCF010001 (allocation number 1947.1015) and the Department of Earth Sciences at Uppsala University. Financial support from the Swedish Research Council and contributing research institutes to the Swedish Integrated Carbon Observation System (ICOS-Sweden) research infrastructure and the Swedish Infrastructure for Ecosystem Science (SITES) are also acknowledged. The study further benefitted from support by the Knut and Alice Wallenberg foundation. We also thank Joachim Audet and the Krycklan Catchment Study crew for field support.

## Author contributions

A.C., M.B.W., M.F.B and K.B. designed the study. M.B.W., K.B. and M.F.B. contributed materials and funding, and H.L. provided infrastructure for the data collection. A.C. and M.B.W. wrote the paper. A.C., M.F.B., M.B.W. and K.B. carried out the fieldwork. M.H.G. helped coordinate the radiocarbon sampling and laboratory analysis. A.C. processed and analysed the data. N.A. modelled the hillslope hydrological export. H.L., M.Ö., M.H.G. and M.F.B. provided scientific insight to the analysis and interpretation of the data. All authors commented on earlier versions of this paper.

## Additional information

**Competing interests:** The authors declare no competing interests.

