## [Peer Review File · Nature Communications]

Reviewers' comments:

Reviewer #1 (Remarks to the Author):

Check grammar throughout: e.g.:

Line 9: Stream CO₂ emissions contribute significantly to the atmospheric climate forcing.

Line 31: Explicit demonstration of this link are still absent.

Line 35-40: no refs for the statements here. Need adding.

The paper provides novel data about the radiocarbon age of DOC and CO₂ in groundwater to stream water for a catchment where there is no carbonate source of CO₂. CO₂ 14C-age is found to be constant and close to contemporary, while DOC is older (either carrying a bomb signal or an even older, depleted signal). This is an exciting finding that reveals a disconnect in the timescales over which these two C pools cycle. The differences in the ages of these two C pools, also indicates that the bulk DOC in the water cannot be the main source of the CO₂. As the CO₂ is modern, then it must come from a modern source. So far, I agree with the authors interpretations and conclusions. However, the authors are very precise in their identification of the modern source of CO₂ – perhaps too precise.

The authors setup a dichotomy between photosynthates from plant roots and DOC as the potential sources of CO₂ for respiration. As the photosynthates (which are not measured in the study) are thought to be rapidly transferred to the soil microbial community and respired, they are postulated as the source of the contemporary CO₂ eventually observed in streams. It is unclear to me whether this system really only has two components. It seems that organics leaching directly from vegetation to soils in throughfall and stem flow (no age data but recent review: Van Stan and Stubbins L&O Letters), the organics in and leaching from leaf litter, and maybe some other physical paths of entry for contemporary particulate or dissolved organic C to the soil, would all provide modern organic C directly to the soil microbes. The authors should consider these other sources of contemporary OC and, if they cannot rule them out explicitly, then they should alter their dichotomy from bulk, aged DOC vs photosynthate to bulk, aged DOC vs contemporary OC ultimately derived from the local vegetation. This to me would not change the thrust or the impact of their findings, but would give a more nuanced view of how the modern OC that they suggest drives CO₂ reaches the soil to be respired.

The authors should also consider that the bulk DOC 14C values are only bulk values. It may be that when DOC is in excess of CO₂, such that only modest DOC would need to be respired, the modern CO₂ could indeed be derived from a modern fraction of the DOC. I'm most suggesting this needs doing for the current paper, but the way to test this would be to conduct biodegradation experiments with the DOC and assess age vs degradation. A similar approach (Mann et al 2015 – Nat Comm) revealed that the DOC respired by microbes in permafrost impacted systems was ancient (depleted in 14C) even when bulk DOC was modern – i.e. there was a small, but highly labile fraction of radiocarbon depleted DOC present in the bulk DOC that the microbes targeted. I could envision the opposite is driving the trends here – a fraction of modern, biolabile DOC is hidden within the bulk 14C signature. This is just part of the caveat to the above and again does not change the main thrust of the paper: i.e. that very recently produced OC is fueling CO₂ – again it is just that it does not seem as simple as the dichotomy the authors present and use to conclude that the CO₂ source has to be the explicitly named photosynthate.

These critiques are significant in that they should be addressed in the manuscript, but they do not detract from the novelty and impact of the paper if addressed. The paper would make a strong addition to Nat Comm once these issues are addressed.

Sincerely,
Aron Stubbins

Reviewer #2 (Remarks to the Author):

This work investigated radiocarbon compositions of stream water CO₂ and dissolved organic carbon (DOC) in Swedish streams. The authors found that stream CO₂ is predominantly derived from young organic matter mineralization in soils. Although the authors argued that this work is the first evidence, I think that a similar implication has been already made by previous studies (after Mayorga et al. 2005 Nature). More importantly, I felt quite incongruous with the term "root respiration". Did the authors directly measure the root respiration in this study? As far as I examined, this statement is only based on the observation that ¹⁴C of stream water CO₂ was similar with that of soil CO₂. This is highly speculative. The ¹⁴C content of stream water CO₂ in the present study can be also explained by a simple mixing of ¹⁴C of DOC and ¹⁴C of atmospheric CO₂. There are indeed some studies that support this scenario (e.g., Ishikawa et al. 2015 Radiocarbon). The delta ¹³C of stream water CO₂ (Fig. S2) might also support this hypothesis. The observed delta ¹³C range (-25 to -20 permil) was distinct from DOC and was too high for terrestrial C₃ vegetation. I would consider contributions from the atmospheric CO₂ and/or in-stream production. If the authors could not refute the above, a drastic revision of the manuscript would be unavoidable.

Reviewer #3 (Remarks to the Author):

Review for manuscript number NCOMMS-18-7748888
Current root respiration fuels stream CO₂ emissions

This paper presents patterns in ¹⁴C-CO₂ and ¹⁴C-DOC from a boreal, headwater stream as well as riparian and upslope groundwaters over one year. Given that the CO₂ and DOC radiocarbon signatures generally showed little overlap, and that the ¹⁴C-CO₂ in the stream was relatively stable and close to the ¹⁴C content in the atmosphere, the authors infer that carbon originally fixed in the terrestrial ecosystem within the past year, specifically root respiration (autotrophic root respiration as well as mineralization of root exudates in soils), is the predominant source of CO₂ to the study stream. Although lateral export of CO₂ could account for less than 5% of annual catchment net ecosystem exchange, the findings from this study suggest that stream CO₂ emissions will be sensitive to disturbances that affect fast-cycling C processes on land.

The goal of this paper – to partition the sources of stream CO₂ – is an emerging area of interest to a broad community of hydrologists, biogeochemists, and earth systems modelers. It is not surprising that groundwater has a disproportionate influence on CO₂ dynamics in a very small stream such as this one (upstream area = 0.13 km²). However, as the authors point out, most studies thus far have relied on mass-balance techniques (and many simplifying assumptions) to partition CO₂ sources in streams, and so the specific biogeochemical processes and mechanisms controlling CO₂ inputs from land, as well as their characteristic timescales, are not well-constrained. The spatial and temporal resolution of this study are limited, but the findings presented here are very interesting and will hopefully influence further mechanistic studies of freshwater CO₂ sources.

Overall, the findings are convincing and the paper is well-written. My main comment is that the discussion section could be made stronger by more explicitly outlining how the radiocarbon and chemistry data inform interpretation of flowpath dynamics, and how generalizable such patterns might be, even within the larger catchment. More specifically:

1) As written, lines 233-238 are repetitive with the results and lack broader interpretation. The authors state that hydrologic retention and flowpaths were the main drivers of ¹⁴C-DOC (but not

14C-CO₂), but the onus seems to be on the reader to synthesize the various spatial and temporal data. Lines 243-245 provide an effective structure for the bulk of the paragraph, but I would have liked to see more interpretation of the results shown in figures 1-4, including description of the dominant flowpaths that supply DOC and CO₂ to the stream, how those flowpaths varied over the year-long study, and the implications of hydrologic connectivity for the nature or age of DOC and CO₂ exported from the hillslope and riparian zone to the stream (e.g. What does it tell us that superficial water tables corresponded to more enriched 14C-DOC in the stream; Figure 1b? Why are 14C-DOC and 14C-CO₂ most divergent during the spring freshet; Figure 1a? Is it significant that 14C-DOC and 14C-CO₂ matched most closely and were least variable across the 3 sampling dates in the deep riparian groundwater; Figure 2?)

2) Within the above discussion, I wonder if the authors could assess the potential generality of their findings, specifically as it relates to the dominant flowpaths that would deliver organic and inorganic C of varying ages to receiving streams. The Krycklan catchment is very well-studied. To what extent is this transect representative of groundwater dynamics in the catchment (e.g. Ledesma et al. 2015), and how might flowpath heterogeneity influence catchment carbon sources or their delivery to streams?

Specific comments by line:

Line 80: "also" should be omitted here.

Line 81: This paragraph is important, since the mixing of C sources of different ages represents a challenge for interpreting mean 14C signatures in bulk samples. I was initially unclear whether the information contained in parentheses (14C >100%) represents a specific assumption being made in this study, although lines 83-84 suggest that the authors are inferring sources simply by differences in 14C content among samples.

Line 115: Indicate that the directionality of the net difference flipped, however, for these deep groundwater samples.

Line 154: Should panel 4f be cited here as reference to the upslope groundwater CO₂ concentrations?

Line 166: There may be no statistical difference in DOC concentrations between the shallow and deeper riparian groundwaters, but Figure 4e suggests that these concentrations diverge considerably throughout the growing season (~30 mg C/L). Noting percent differences in concentration at the beginning and end of the sampling may be useful for highlighting this dynamic.

Line 181: I had trouble interpreting the relative contributions of runoff from shallow depths in Figure S1. Indicating which panel shows this finding would be helpful.

Line 242: "these patterns" is vague as the subject in this sentence. It is better to make clear that "Instead, the consistent separation of 14C-DOC and 14C-CO₂ in streams and groundwater indicate..." or another variation on this clause that could help point the reader to the specific data supporting the claims of "millennial timescales" (I assume this is inferred from the deep upslope and riparian groundwater samples that were highly depleted in August and October) and DOC arising from "more diverse sources."

Line 249: Figure 4 appears to show DOC concentrations rather than 14C-DOC values. It also seems worth articulating that the differences in 14C-DOC between riparian and upslope groundwaters were largely driven by dynamics in deeper flow paths later in the growing season.

Line 260: This is an important finding, but appears for the first time here (results from the mass balance calculations do not appear in section 3.3 of the Results as far as I can tell). Does this 66% figure come from the calculation of R_r (line 478)?

Lines 279-281. I found the lack of seasonality in ^{14}C - CO_2 (and presumably R_r , the ratio of stream CO_2 that could be explained by root respiration) to be very surprising. Perhaps the lack of clear seasonality is a result of the limited discrete temporal sampling, but it is interesting that the pool of CO_2 derived from root respiration is potentially large enough to sustain lateral CO_2 fluxes throughout the year.

Line 292: Please specify which conditions other than water residence time are being referred to here (with respect to large rivers).

Line 293: I initially found this line of reasoning unclear, and would suggest rephrasing. Perhaps, "...there is a possibility that the ^{14}C -DOC in this catchment may reflect refractory DOC compounds not mineralized during transit, masking an underlying connection between ^{14}C - CO_2 and ^{14}C -DOC."

Line 299: Does "these C substrates" refer to aged sources of DOC, from the deep upslope groundwater, for example?

Line 307: Can this claim that lateral CO_2 export remobilizes a significant fraction of the C originally fixed on land be reasonably supported given that lateral CO_2 export represents 2% of annual catchment NEE in this study (paragraph starting on line 254)?

Lines 311-313: But why would greater root activity in tropical and temperate forests increase the root contribution to lateral CO_2 export, if, in this study, it does not appear to be the size of the root-derived CO_2 pool that limits lateral CO_2 flux, but rather, hydrologic connectivity? Could catchment topographic controls, or the distribution of preferential flow paths created by root macropores be more important than biome for considering the generality of these findings?

Line 377: I see that the distances between the groundwater wells and the stream are reported, but could the authors also provide any estimates of mean travel time to the stream (i.e. if available from prior research in this catchment)? For example, is it reasonable to assume that CO_2 derived from recent (<1 year) root activity in the upslope groundwater would travel to the stream during this timeframe?

Figure 3: Note in the caption that the two excluded outlier samples are denoted with an asterisk in the figure. Please also clarify the meaning of the light gray line. I assume this is meant to represent current atmospheric ^{14}C content, but I am not sure.

Figure 5. It would be helpful to clarify in the figure caption that the two arrows pointing towards the stream benthos in the bottom panel represent downstream fluxes.

Figure S2: This figure showing the ^{14}C - ^{13}C biplot is not referenced in the text, but the data seem useful for arguing against a modern stream CO_2 signature that arises simply from reaeration, for example.

#	Lines	Reviewer 1 comments:	Author's Response
1		The paper provides novel data about the radiocarbon age of DOC and CO₂ in groundwater to stream water for a catchment where there is no carbonate source of CO₂. CO₂ 14C-age is found to be constant and close to contemporary, while DOC is older (either carrying a bomb signal or an even older, depleted signal). This is an exciting finding that reveals a disconnect in the timescales over which these two C pools cycle. The differences in the ages of these two C pools, also indicates that the bulk DOC in the water cannot be the main source of the CO₂. As the CO₂ is modern, then it must come from a modern source. So far, I agree with the authors interpretations and conclusions. However, the authors are very precise in their identification of the modern source of CO₂ – perhaps too precise. The authors setup a dichotomy between photosynthates from plant roots and DOC as the potential sources of CO₂ for respiration. As the photosynthates (which are not measured in the study) are thought to be rapidly transferred to the soil microbial community and respired, they are postulated as the source of the contemporary CO₂ eventually observed in streams. It is unclear to me whether this system really only has two components. It seems that organics leaching directly from vegetation to soils in throughfall and stem flow (no age data but recent review: Van Stan and Stubbins L&O Letters), the organics in and leaching from leaf litter, and maybe some other physical paths of entry for contemporary particulate or dissolved organic C to the soil, would all provide modern organic C directly to the soil microbes. The authors should consider these other sources of contemporary OC and, if they cannot rule them out explicitly, then they should alter their dichotomy from bulk, aged DOC vs photosynthate to bulk, aged DOC vs contemporary OC ultimately derived from the local vegetation. This to me would not change the thrust or the impact of their findings, but would give a more nuanced view of how the modern OC that they suggest drives CO₂ reaches the soil to be respired.	We thank Dr. Stubbins for his positive comments and valuable recommendations on the manuscript. An important concern was raised with regards to the definitions of the processes separated in the study. Since root respired ¹⁴C-CO₂ was not measured in the study, we agree that our original formulation was perhaps too precise or speculative. Root associated respiration is arguably one of the key process that generates soil CO₂ from recent photosynthates, thus supporting the close agreement with the current atmospheric level, but as pointed out, it may not be the only process. We changed the formulation of the CO₂ sources separated here from “current root respiration” to “current forest C fixation”. With this new formulation, we avoid being too precise with regards to the biological processes and C allocation patterns occurring in the catchment - not measured in the study. The term forest C fixation encompass all biological processes that results in the incorporation of CO₂ with a ¹⁴C content reflecting the current atmospheric level in the catchment. The title of the manuscript has been changed according to this new formulation (Lines 1). In the introduction (Lines 40-58) and the discussion (throughout), we describe the biological processes that could result in the production of soil and groundwater CO₂ with a ¹⁴C content reflecting the atmospheric CO₂ fixed from forest vegetation during the study year. In detailing these processes, we further refer to the paper mentioned by Dr. Stubbins on DOC transport via throughfall and stemflow (Van Stan and Stubbins L&O Letters, 2018), which was not included in the original version of the manuscript. Van Stan, J. T., and A. Stubbins (2018), Tree-DOM: Dissolved organic matter in throughfall and stemflow, Limnology and Oceanography Letters, 3(3), 199-214, doi:10.1002/lol2.10059.
3		The authors should also consider that the bulk DOC 14C values are only bulk values. It may be that when DOC is in excess of CO₂, such that only modest DOC would need to be respired, the modern CO₂ could indeed be derived from a modern fraction of the DOC. I'm most suggesting this needs doing for the current paper, but the way to test this would be to conduct biodegradation experiments with the DOC and assess age vs degradation. A similar approach	By changing the formulation of the two main CO₂ sources separated in the manuscript (comment#1), we can now include this possibility in our assessment of CO₂ sources. As stated by Dr. Stubbins, the close agreement between groundwater and stream water ¹⁴C-CO₂ and the current atmospheric ¹⁴C level could arise from the selective decay of recent fractions of the DOC pool, which is not clearly represented in the bulk ¹⁴C content value. This

	(Mann et al 2015 – Nat Comm) revealed that the DOC respired by microbes in permafrost impacted systems was ancient (depleted in 14C) even when bulk DOC was modern – i.e. there was a small, but highly labile fraction of radiocarbon depleted DOC present in the bulk DOC that the microbes targeted. I could envision the opposite is driving the trends here – a fraction of modern, biolabile DOC is hidden within the bulk 14C signature. This is just part of the caveat to the above and again does not change the main thrust of the paper: i.e. that very recently produced OC is fueling CO2 – again it is just that it does not seem as simple as the dichotomy the authors present and use to conclude that the CO2 source has to be the explicitly named photosynthate.	possibility is now explicitly stated (Lines 282-284). Moreover, we now refer to Mann et al., (2018) and Dean et al.,(2019) in the discussion section (Lines 302-304), stating the potential for selective decay of DOC, which may not represent the bulk. Dean, J. F., Garnett, M. H., Spyrakos, E. and Billett, M. F.: The potential hidden age of dissolved organic carbon exported by peatland streams, J. Geophys. Res. Biogeosciences, doi:10.1029/2018JG004650, 2019. Mann, P. J., Eglinton, T. I., McIntyre, C. P., Zimov, N., Davydova, A., Vonk, J. E., Holmes, R. M. and Spencer, R. G. M.: Utilization of ancient permafrost carbon in headwaters of Arctic fluvial networks, Nat Commun., 6, doi:10.1038/ncomms8856, 2015. The potential for DOC mineralisation within the catchment stream waters is also discussed in more detail (Lines 305-307), where we refer to the various studies that demonstrated DOC degradation across the catchment’s surface waters to be low (see references below): Winterdahl, M., Wallin, M. B., Karlsen, R. H., Laudon, H., Öquist, M. and Lyon, S. W.: Decoupling of carbon dioxide and dissolved organic carbon in boreal headwater streams, J. Geophys. Res. Biogeosciences, 121(10), 2630–2651, doi:10.1002/2016jg003420, 2016. Kothawala, D. N., Ji, X., Laudon, H., Ågren, A. M., Futter, M. N., Köhler, S. J. and Tranvik, L. J.: The relative influence of land cover, hydrology, and in-stream processing on the composition of dissolved organic matter in boreal streams, J. Geophys. Res. Biogeosciences, n/a-n/a, doi:10.1002/2015JG002946, 2015. Köhler S, Buffam I, Jonsson A, Bishop K. Photochemical and microbial processing of stream and soil water dissolved organic matter in a boreal forested catchment in northern Sweden. Aquatic Sciences 64, 269-281 (2002).
4	These critiques are significant in that they should be addressed in the manuscript, but they do not detract from the novelty and impact of the paper if addressed. The paper would make a strong addition to Nat Comm once these issues are addressed.	We thank Dr. Stubbins for raising these points as we find that the manuscript has been considerably improved by addressing these issues.

5	9	Check gramma : “Stream CO2 emissions contribute significantly to the atmospheric climate forcing.”	Corrected (Lines 9)
6	31	Check gramma: “Explicit demonstration of this link are still absent”	Corrected (Lines 32)
7	35-40	no refs for the statements here. Need adding.	This section details the processes that are separated in the study. Following the reviewer’s previous comments, this section has been modified substantially (Lines 37-55). We have added the following references to these definitions: Kuzyakov, Y.: Sources of CO2 efflux from soil and review of partitioning methods, Soil Biol. Biochem., 38(3), 425–448, doi:10.1016/j.soilbio.2005.08.020, 2006. Schuur, E. A. G. & Trumbore, S. E. Partitioning sources of soil respiration in boreal black spruce forest using radiocarbon. Glob. Chang. Biol. 12, 165–176 (2006). Hanson, P. J., Edwards, N. T., Garten, C. T. & Andrews, J. A. Separating root and soil microbial contributions to soil respiration: A review of methods and observations. Biogeochemistry 48, 115–146 (2000). Hahn, V., Högberg, P. & Buchmann, N. 14C - A tool for separation of autotrophic and heterotrophic soil respiration. Glob. Chang. Biol. 12, 972–982 (2006). Trumbore, S. (2000), Age of soil organic matter and soil respiration: radiocarbon constraints on belowground C dynamics, Ecological Applications, 10, 399-411, doi:10.1890/1051-0761(2000)010[0399:AOSOMA]2.0.CO;2. Van Stan, J. T. and Stubbins, A.: Tree-DOM: Dissolved organic matter in throughfall and stemflow, Limnol. Oceanogr. Lett., 199–214, doi:10.1002/lo2.10059, 2018.

#	Lines	Reviewer 2 comments:	Author's Response
1		This work investigated radiocarbon compositions of stream water CO₂ and dissolved organic carbon (DOC) in Swedish streams. The authors found that stream CO₂ is predominantly derived from young organic matter mineralization in soils. Although the authors argued that this work is the first evidence, I think that a similar implication has been already made by previous studies (after Mayorga et al. 2005 Nature).	We thank the reviewer for pointing out this aspect that was not clearly stated in the original version of this manuscript. The reviewer is right that our findings concur with those of Mayorga et al., (2005), in the sense that:  1) a number of ¹⁴C-CO₂ values reported from the Amazon river basin are close to the current atmospheric level, as in our study. 2) important contrasts between the ¹⁴C-DOC and ¹⁴C-CO₂ were also observed in this catchment. These two aspects are now explicitly stated in the manuscript (Point 1: 234-236, Point 2: Lines 224-226). Still, major differences between the two studies distinguishes our findings from those of Mayorga et al., (2005). The CO₂ pool in the Amazon river is derived from three different sources, C3 and C4 plant, and weathering of carbonate containing minerals. This greatly complicates the interpretation of CO₂ sources for the Amazon river basin by δ¹³C and ¹⁴C characterisation. For example, in the Amazon river, the ¹⁴C-CO₂ varies from 25 to 114 %modern, and δ¹³C-CO₂ varies from -26.3 -10.7 ‰, which is far more variable than in our study (¹⁴C-CO₂ = 102 to 105, δ¹³C-CO₂ = -25.1 to -20.4‰) The CO₂ sources in our catchment are simpler in comparison with the Amazon river basin, since they exclude C4 plant and weathering of carbonate containing minerals (Lines 232-233). This allowed us to go further in the separation of CO₂ sources, whereby we could directly compare the ¹⁴C content of DOC and CO₂ in order to separate the CO₂ sources, which was not possible for Mayorga et al., (2005). In addition, the connection to terrestrial sources was not explicit in Mayorga et al, 2005. Their study reported ¹⁴C content in various C species from rivers and streams, but did not include groundwater sources. In fact, Mayorga et al., (2005), claims that most of the CO₂ supersaturation across the Amazon river basin derived from in-situ aquatic respiration, rather than from groundwater sources, which again differs from our findings. In our study, both groundwater and stream water ¹⁴C-DOC and ¹⁴C-CO₂ was characterised, which allowed us

			to explicitly demonstrate that stream CO₂ are derived from terrestrial sources. To better illustrate the differences and similarities between the results of the two studies, we now overlay the data from Mayorga et al., (2005) on the scatterplot showing ¹⁴C against δ¹³C values with ours in Supplementary Figure 2. Mayorga, E., A. K. Aufdenkampe, C. A. Masiello, A. V. Krusche, J. I. Hedges, P. Quay, J. E. Richey, and T. A. Brown (2005), Young organic matter as a source of carbon dioxide outgassing from Amazonian rivers, Nature, 436(7050), 538-541, doi:10.1038/nature03880.
2		More importantly, I felt quite incongruous with the term “root respiration”. Did the authors directly measure the root respiration in this study? As far as I examined, this statement is only based on the observation that ¹⁴C of stream water CO₂ was similar with that of soil CO₂. This is highly speculative.	We changed the terminology according to the comments from Reviewer 1 (see comment #1). It is, however, worth noting that this assumption, i.e. root respired CO₂ is in tune with the current atmospheric ¹⁴C level, is common across the literature, for example in the references below. In the event of an older storage contributing to root respired CO₂, for example as reported Cisneros-Dozal et al., (2006) up to 5yrs), the relative contribution of the root respiration would increase, making our estimate conservative. Schuur, E. A. G. & Trumbore, S. E. Partitioning sources of soil respiration in boreal black spruce forest using radiocarbon. Glob. Chang. Biol. 12, 165–176 (2006) Hanson, P. J., Edwards, N. T., Garten, C. T. & Andrews, J. A. Separating root and soil microbial contributions to soil respiration: A review of methods and observations. Biogeochemistry 48, 115–146 (2000). Hahn, V., Högberg, P. & Buchmann, N. ¹⁴C - A tool for separation of autotrophic and heterotrophic soil respiration. Glob. Chang. Biol. 12, 972–982 (2006). Cisneros-Dozal, L. M., Trumbore, S., and Hanson, P. J. (2006), Partitioning sources of soil-respired CO₂ and their seasonal variation using a unique radiocarbon tracer, Global Change Biology, 12, 194-204, doi:doi:10.1111/j.1365-2486.2005.001061.x.)

3	The $\delta^{14}\text{C}$ content of stream water CO_2 in the present study can be also explained by a simple mixing of $\delta^{14}\text{C}$ of DOC and $\delta^{14}\text{C}$ of atmospheric CO_2. There are indeed some studies that support this scenario (e.g., Ishikawa et al. 2015 Radiocarbon). The $\delta^{13}\text{C}$ of stream water CO_2 (Fig. S2) might also support this hypothesis. The observed $\delta^{13}\text{C}$ range (-25 to -20 permil) was distinct from DOC and was too high for terrestrial C_3 vegetation. I would consider contributions from the atmospheric CO_2 and/or in-stream production. If the authors could not refute the above, a drastic revision of the manuscript would be unavoidable.	The patterns in $\delta^{13}\text{C}$-CO_2 values in the groundwater and stream water at the same site have been described in detail in Campeau et al (2018) (see full ref below). The $\delta^{13}\text{C}$-CO_2 values in groundwater and stream water are not affected by atmospheric CO_2 invasion (Lines 245-249). The CO_2 supersaturation are always far larger than the atmosphere, as indicated by the hourly CO_2 concentration measurements presented in Figures 4d-f and thus CO_2 systematically evades from the groundwater and stream water toward the atmosphere – not the other way (Lines 245-249) The $\delta^{13}\text{C}$-CO_2 values are more positive than the $\delta^{13}\text{C}$-DOC because of kinetic fractionation during CO_2 evasion to the atmosphere (again not invasion) and also because of methanogenesis in the riparian groundwater (Campeau et al, 2018). Stream productivity is also very low at this site (Berggren et al., 2007, Berggren et al., 2009), thus $\delta^{13}\text{C}$-CO_2 values are not significantly affected by stream photosynthesis, as indicated by the similarity between the stream water and groundwaters (Campeau et al, 2018). Campeau, A., Bishop, K., Nilsson, M. B., Klemetsson, L., Laudon, H., Leith, F. I., Oquist, M. and Wallin, M. B.: Stable Carbon Isotopes Reveal Soil-Stream DIC Linkages in Contrasting Headwater Catchments, J. Geophys. Res., 123(1), 149–167, doi:10.1002/2017jg004083, 2018. Berggren M , Laudon H and Jansson M 2007. Landscape regulation of bacterial growth efficiency in boreal freshwater. Global Biogeochemical Cycles21: Art. No. GB4002. DOI: 10.1029 / 2006GB002844 Berggren M , Laudon H and Jansson M. 2009. Hydrological control of organic carbon support for bacterial growth in boreal headwater streams. Microbial Ecology 57 (1): 170-178. DOI: 10.1007 / s00248-008-9423-6
4		

#	Lines	Reviewer 3 comments:	Author's Response
1		This paper presents patterns in 14C-CO2 and 14C-DOC from a boreal, headwater stream as well as riparian and upslope groundwaters over one year. Given that the CO2 and DOC radiocarbon signatures generally showed little overlap, and that the 14C-CO2 in the stream was relatively stable and close to the 14C content in the atmosphere, the authors infer that carbon originally fixed in the terrestrial ecosystem within the past year, specifically root respiration (autotrophic root respiration as well as mineralization of root exudates in soils), is the predominant source of CO2 to the study stream. Although lateral export of CO2 could account for less than 5% of annual catchment net ecosystem exchange, the findings from this study suggest that stream CO2 emissions will be sensitive to disturbances that affect fast-cycling C processes on land. The goal of this paper – to partition the sources of stream CO2 – is an emerging area of interest to a broad community of hydrologists, biogeochemists, and earth systems modelers. It is not surprising that groundwater has a disproportionate influence on CO2 dynamics in a very small stream such as this one (upstream area = 0.13 km²). However, as the authors point out, most studies thus far have relied on mass-balance techniques (and many simplifying assumptions) to partition CO2 sources in streams, and so the specific biogeochemical processes and mechanisms controlling CO2 inputs from land, as well as their characteristic timescales, are not well-constrained. The spatial and temporal resolution of this study are limited, but the findings presented here are very interesting and will hopefully influence further mechanistic studies of freshwater CO2 sources. Overall, the findings are convincing and the paper is well-written. My main comment is that the discussion section could be made stronger by more explicitly outlining how the radiocarbon and chemistry data inform interpretation of flowpath dynamics, and how generalizable such patterns might be, even within the larger catchment. More specifically:	We thank the reviewer for his/her positive response and detailed revision of our manuscript. Several recommendations were made by the reviewer, which we took in full considerations. We find that these recommendations have substantially improved the manuscript.
2		1- As written, lines 233-238 are repetitive with the results and lack broader interpretation. The authors state that hydrologic retention and flowpaths were the main drivers of 14C-DOC (but not 14C-CO2), but the onus seems to be on the reader to synthesize the various spatial and temporal data. Lines 243-245 provide an effective structure for the bulk of the paragraph, but I would have liked to see	We thank the reviewer for these recommendations and have addressed each of these points below: Point 1-3 are addressed on Lines: 221-222, we describe the hydrological flowpaths in connection with our results (e.g. spring freshet export 14C enriched DOC, upslope groundwater export 14C-depleted DOC). The following references were also added to support our interpretation of the

	more interpretation of the results shown in figures 1-4, including  2- description of the dominant flowpaths that supply DOC and CO₂ to the stream, how those flowpaths varied over the year-long study, 3- and the implications of hydrologic connectivity for the nature or age of DOC and CO₂ exported from the hillslope and riparian zone to the stream (e.g. What does it tell us that superficial water tables corresponded to more enriched ¹⁴C-DOC in the stream; Figure 1b? 4- Why are ¹⁴C-DOC and ¹⁴C-CO₂ most divergent during the spring freshet; Figure 1a? 5- Is it significant that ¹⁴C-DOC and ¹⁴C-CO₂ matched most closely and were least variable across the 3 sampling dates in the deep riparian groundwater; Figure 2?) 	data: Ledesma, J. I. J., Grabs, T., Bishop, K. H., Schiff, S. L. and Köhler, S. J.: Potential for long term transfer of DOC from riparian zones to streams in boreal catchments, Glob. Chang. Biol. [online] Available from: internal-pdf://228.60.152.53/Ledesma-2015-Potential for long-term transfer.pdf, 2015. Köhler SJ, Buffam I, Seibert J, Bishop KH, Laudon H. Dynamics of stream water TOC concentrations in a boreal headwater catchment: Controlling factors and implications for climate scenarios. Journal of Hydrology 373, 44-56 (2009). Pacific VJ, Jencso KG, McGlynn BL. Variable flushing mechanisms and landscape structure control stream DOC export during snowmelt in a set of nested catchments. Biogeochemistry 99, 193-211 (2010). Peralta-Tapia, A., R. A. Sponseller, D. Tetzlaff, C. Soulsby, and H. Laudon (2015), Connecting precipitation inputs and soil flow pathways to stream water in contrasting boreal catchments, Hydrological Processes, 29, 3546-3555, doi:10.1002/hyp.10300. Ameli AA, McDonnell JJ, Bishop K. The exponential decline in saturated hydraulic conductivity with depth: a novel method for exploring its effect on water flow paths and transit time distribution. Hydrological Processes 30, 2438-2450 (2016). Ameli AA, et al. Hillslope permeability architecture controls on subsurface transit time distribution and flow paths. Journal of Hydrology 543, 17-30 (2016). Amvrosiadi N, Seibert J, Grabs T, Bishop K. Water storage dynamics in a till hillslope: the foundation for modeling flows and turnover times. Hydrological Processes 31, 4-14 (2017). Point 4. This point is now discussed explicitly in Lines 208-211 Point 5: This point is now discussed explicitly in Lines 288-293
3	2) Within the above discussion, I wonder if the authors could assess the potential generality of their findings, specifically as it relates to the dominant	We agree with the reviewer that the manuscript could benefit from an assessment of the generality of our findings. Since ¹⁴C-CO₂ showed no

		flowpaths that would deliver organic and inorganic C of varying ages to receiving streams. The Krycklan catchment is very well-studied. To what extent is this transect representative of groundwater dynamics in the catchment (e.g. Ledesma et al. 2015), and how might flowpath heterogeneity influence catchment carbon sources or their delivery to streams?	connection to hydrological drivers (now explicitly discussed on Lines 214-222) nor to any measured variable in the catchment (Lines 249-258), we did not find it would be appropriate to generalise our findings on these basis. Instead, we claim that the processes associated with rapid cycling of soil CO₂ are widespread in forested ecosystems (Lines 320-322) and groundwater are considered as a main source of CO₂ to streams (Lines 317-320) and support these with other published data on ¹⁴C-CO₂ in surface waters (Lines 236-238), and studies on the partitioning of vertical soil CO₂ efflux (Lines 263-266). In a different section, we discuss the changes in hydrological flowpaths across the Krycklan catchment and the possible implications on C cycling (Line 303-309), based on the following references: Tiwari, T., Laudon, H., Beven, K. and Ågren, A. M.: Downstream changes in DOC: Inferring contributions in the face of model uncertainties, Water Resour. Res., 50(1), 514–525, doi:10.1002/2013WR014275, 2014. Peralta-Tapia, A., Ågren, A., Laudon, H., Sponseller, R. A., Tetzlaff, D. and Soulsby, C.: Scale-dependent groundwater contributions influence patterns of winter baseflow stream chemistry in boreal catchments, J. Geophys. Res. Biogeosciences, 2014JG002878, doi:10.1002/2014JG002878, 2015.
4	80	“also” should be omitted here.	Corrected
5	81	This paragraph is important, since the mixing of C sources of different ages represents a challenge for interpreting mean 14C signatures in bulk samples. I was initially unclear whether the information contained in parentheses (14C >100%) represents a specific assumption being made in this study, although lines 83-84 suggest that the authors are inferring sources simply by differences in 14C content among samples.	We clarified our statement (Lines 81-83)
6	115	Indicate that the directionality of the net difference flipped, however, for these deep groundwater samples.	Corrected (Line 120)
7	154	Should panel 4f be cited here as reference to the upslope groundwater CO ₂ concentrations?	Now added reference to Figures 4 e and f
8	166	There may be no statistical difference in DOC concentrations between the shallow and deeper riparian groundwaters, but Figure 4e suggests that these concentrations diverge considerably throughout the growing season (~30 mg C/L). Noting percent differences in concentration at the beginning and end of	Now stated % increase throughout the growing season (Lines 146-148).

		the sampling may be useful for highlighting this dynamic.	
9	181	I had trouble interpreting the relative contributions of runoff from shallow depths in Figure S1. Indicating which panel shows this finding would be helpful.	Reference to specific figure panels now added (Line 164)
10	242	“these patterns” is vague as the subject in this sentence. It is better to make clear that “Instead, the consistent separation of 14C-DOC and 14C-CO ₂ in streams and groundwater indicate...” or another variation on this clause that could help point the reader to the specific data supporting the claims of “millennial timescales” (I assume this is inferred from the deep upslope and riparian groundwater samples that were highly depleted in August and October) and DOC arising from “more diverse sources.”	Now corrected using the reviewer’s suggested wording. (Lines 201-206)
11	249	Figure 4 appears to show DOC concentrations rather than 14C-DOC values. It also seems worth articulating that the differences in 14C-DOC between riparian and upslope groundwaters were largely driven by dynamics in deeper flow paths later in the growing season.	Corrected. The deeper flowpath contribution is now emphasized in the discussion section (Lines 211-214) as stated in our reply to the comment 2.
12	260	This is an important finding, but appears for the first time here (results from the mass balance calculations do not appear in section 3.3 of the Results as far as I can tell). Does this 66% figure come from the calculation of R _r (line 478)?	The results from these calculations are stated on Lines 173-177 . We also revised and simplified those calculations to be only based on the cumulative ¹⁴ C content of the annual lateral C export from the catchment (Figure 5), the contribution of currently fixed CO ₂ from the atmosphere in the lateral CO ₂ export could be up to 75%.
13	279-281	I found the lack of seasonality in 14C-CO ₂ (and presumably R _r , the ratio of stream CO ₂ that could be explained by root respiration) to be very surprising. Perhaps the lack of clear seasonality is a result of the limited discrete temporal sampling, but it is interesting that the pool of CO ₂ derived from root respiration is potentially large enough to sustain lateral CO ₂ fluxes throughout the year.	The lack of seasonality is further emphasized in connection with the changes in hydrological conditions that appeared to affect the ¹⁴ C-DOC, but not the ¹⁴ C-CO ₂ . As stated in the manuscript, Lines 214-222 , the amount of CO ₂ fixed from the atmosphere during the growing season was likely sufficient to sustain the lateral export of CO ₂ throughout the year. Yet the biological pathways transporting and mineralizing this into soils can shift throughout seasons, without changing the age signature (This is now explicitly stated in Lines 249-258).
14	292	Please specify which conditions other than water residence time are being referred to here (with respect to large rivers).	We have modified this sentence to make it more specific and replaced “conditions” by “Decreased connectivity with catchment soils” (Lines 295-397)
15	293	I initially found this line of reasoning unclear, and would suggest rephrasing. Perhaps, “...there is a possibility that the 14C-DOC in this catchment may reflect refractory DOC compounds not mineralized during transit, masking an underlying connection between 14C-CO ₂ and 14C-DOC.”	This sentence is now modified following the reviewer’s recommendation. (Lines 281-283)

16	299	Does “these C substrates” refer to aged sources of DOC, from the deep upslope groundwater, for example?	We referred to both the aged-DOC and the DOC with a clear intrusion of post-bomb ¹⁴ C (now clarified in the same statement, Lines 298-301)
17	307	Can this claim that lateral CO ₂ export remobilizes a significant fraction of the C originally fixed on land be reasonably supported given that lateral CO ₂ export represents 2% of annual catchment NEE in this study (paragraph starting on line 254)?	We now specify that based on Öquist et al, 2014, the fraction of NEE exported to the stream as CO ₂ can be up to 9% in other years. Line 279
18	311-313	But why would greater root activity in tropical and temperate forests increase the root contribution to lateral CO ₂ export, if, in this study, it does not appear to be the size of the root-derived CO ₂ pool that limits lateral CO ₂ flux, but rather, hydrologic connectivity? Could catchment topographic controls, or the distribution of preferential flow paths created by root macropores be more important than biome for considering the generality of these findings?	Since the groundwater and stream water ¹⁴C-CO₂ in this catchment showed no connection to hydro-climatic conditions, such generalisation for other catchments is hard to make. Since autotrophic root respiration is known to hold a large contribution to soil CO₂ efflux, and the transport of recent photosynthate to forested soils is also considerable, we claim that these CO₂ sources are likely widespread across forested catchment. (Line 263-266, 320-322) As indicated by the reviewer, the hydrological connection with the landscape possibly decreases further downstream, where aquatic mineralization of terrestrial DOC can supplement these sources (Now stated in Line 294-309).
19	377	I see that the distances between the groundwater wells and the stream are reported, but could the authors also provide any estimates of mean travel time to the stream (i.e. if available from prior research in this catchment)? For example, is it reasonable to assume that CO ₂ derived from recent (<1 year) root activity in the upslope groundwater would travel to the stream during this timeframe?	We now present estimates of the 10 year mean water travel time (Peralta-Tapia et al, 2016) and turnover time for the entire catchment area, as well as an estimate of the time for water to exit the studied 12m transect area (Amvrosiadi et al., 2017) (Lines 388-394)
20	Fig.3	Note in the caption that the two excluded outlier samples are denoted with an asterisk in the figure. Please also clarify the meaning of the light gray line. I assume this is meant to represent current atmospheric ¹⁴ C content, but I am not sure.	The outliers labelling is now clarified in the caption and the grey line is now described with text in the Figure 3
21	Fig.5	It would be helpful to clarify in the figure caption that the two arrows pointing towards the stream benthos in the bottom panel represent downstream fluxes.	Downstream export is now written next to the arrows in Figure 5 , along with CO ₂ emissions and lateral export next to their respective arrows.
22	Fig.S2	This figure showing the ¹⁴ C- ¹³ C biplot is not referenced in the text, but the data seem useful for arguing against a modern stream CO ₂ signature that arises simply from reaeration, for example.	We thank the reviewer for noticing this mistake. We now reference the Supplementary Figure 2 on Lines 280, 288,305 , and have superimposed literature data next to ours, including the Amazon river (Mayorga), and data from various studies in peatland catchments (Leith et al, 2014, Billett et al, 2014, and Campeau et al, 2017) to show the range in published ¹⁴ C and ^δ ¹³ C value for DOC and CO ₂ .

Additional changes:

Line 13-16 were slightly repetitive. We changed the sentence *Lines 13-15* to focus more on the results, and slightly reformulated *Line 15-16* to emphasize the conclusions of the study.

Figure 1a: An error was noticed regarding the point placements on the figures. A number of CO₂ (ti) samples were misplaced along the time axis due to a coding error. A new figure showing the appropriate values is now presented in Figure 1.

A typo mistake was identified and corrected on *Line 163*.

REVIEWERS' COMMENTS:

Reviewer #1 (Remarks to the Author):

As in my original review I find this paper to present compelling, novel information about the disconnect in age and therefore source of DOC and CO₂ in the study system. The authors responded to my previous comments. I believe the paper is now ready for publication and would make an excellent, novel contribution to Nature Communications.

In looking at the other reviews of the paper, I am not as concerned about the overlap with Mayorga et al as one reviewer. The current paper provides novel data about the radiocarbon age of DOC and CO₂ in groundwater to stream water for a catchment where there is no carbonate source of CO₂. This absence of a lack of a third endmember of CO₂ (i.e. groundwater) is critical to the interpretations as I understand them and marks the current study apart from the Mayorga et al study which was based in the Amazon where there are groundwater CO₂ sources.

Reviewer #2 (Remarks to the Author):

The bubble plots in Fig. 2 and Fig. 5 are very strange. This kind of plot is often used to represent quantitative data such as mass, area or volume. These variables have explicit units like kg, km² or ml. On the other hand, %modern is dimensionless and just indicates the deviation of ¹⁴C activity relative to that in a given year (i.e. AD 1950). The authors should refer to Stuiver and Polach (1977) Radiocarbon for detail. I suggest the authors remove the bubbles and instead only show the numbers, preferably as the form of mean +/- standard deviation.

Reviewer #3 (Remarks to the Author):

Review for manuscript number NCOMMS-18-7748888A
Current forest C fixation fuels stream CO₂ emissions

This paper shows that in a boreal, headwater stream, the radiocarbon ages of CO₂ and DOC generally showed little overlap over one year, suggesting that these two carbon pools are primarily sourced from separate biogeochemical pathways that operate over different timescales within the study catchment. Specifically, the ¹⁴C-CO₂ in stream- and groundwater was generally indicative of a contemporary source while ¹⁴C-DOC was more variable and likely originated from an older carbon pool.

This manuscript is an improved version of one I reviewed before. In this revision, the authors have broadened their assignment and interpretation of the modern carbon source feeding streams beyond just root respiration. This change, as well the edits made to the discussion that add interpretation of the flowpaths that deliver CO₂ from catchment soils to streams, have helped to clarify the text and the study findings. I do not have any substantial hesitations regarding the findings or their interpretation, but offer several minor comments below that may further clarify parts of the manuscript presentation.

Specific comments by line:

Line 16: I wonder if "recently fixed" or "contemporary carbon" would be a more fitting word choice throughout the text to replace "current" or "currently fixed." I leave this up to the authors.

Lines 19-20: Stream CO₂ emissions, specifically, will also be highly dependent on patterns of hydrologic variation (e.g. Liu and Raymond, 2019 - Hydrologic controls on pCO₂ and CO₂ efflux in US streams and rivers), so it is difficult to envision what such responses in stream CO₂ emissions

might look like. It may be simpler to state that stream CO₂ fluxes (and/or emissions) will be highly sensitive to patterns of forest C allocation.

Line 41: It is sufficient to say "over longer timescales" here.

Lines 166-167: I am somewhat confused by this statement. Is this comparison between CO₂ export in the upslope and riparian locations made by comparing areal flux rates (as in Figure 5)? If so, comparing these rates in the context of the relative catchment area comprised by riparian soils is not a meaningful comparison compared with export estimates from each zone that are scaled to the whole catchment, for example.

Line 238: Please add more specificity, i.e. rapid C cycling processes within catchment soils.

Line 194: Is "fluxes" the right word here? Consider replacing with "sources."

REVIEWERS' COMMENTS:

Reviewer #1 (Remarks to the Author):

As in my original review I find this paper to present compelling, novel information about the disconnect in age and therefore source of DOC and CO₂ in the study system. The authors responded to my previous comments. I believe the paper is now ready for publication and would make an excellent, novel contribution to Nature Communications.

In looking at the other reviews of the paper, I am not as concerned about the overlap with Mayorga et al as one reviewer. The current paper provides novel data about the radiocarbon age of DOC and CO₂ in groundwater to stream water for a catchment where there is no carbonate source of CO₂. This absence of a lack of a third endmember of CO₂ (i.e. groundwater) is critical to the interpretations as I understand them and marks the current study apart from the Mayorga et al study which was based in the Amazon where there are groundwater CO₂ sources.

We thank the reviewer for this positive recommendation of our manuscript.

Reviewer #2 (Remarks to the Author):

The bubble plots in Fig. 2 and Fig. 5 are very strange. This kind of plot is often used to represent quantitative data such as mass, area or volume. These variables have explicit units like kg, km² or ml. On the other hand, %modern is dimensionless and just indicates the deviation of ¹⁴C activity relative to that in a given year (i.e. AD 1950). The authors should refer to Stuiver and Polach (1977) Radiocarbon for detail. I suggest the authors remove the bubbles and instead only show the numbers, preferably as the form of mean +/- standard deviation.

Since the axis on these plots represent the sample locations, the only way to present the ¹⁴C results in a visually telling way is by altering symbols sizes of the points. By showing only the numbers, as suggested by the reviewer, we would be asking the reader to check each individual values, which would make impossible to get an overall view of the ¹⁴C patterns. The standard deviations are already provided in the supplementary table 1 and 2 with the original data, which would make it redundant information if included in the figure.

While we agree with the reviewer that these bubble plots are often used to report masses, we find that there is no conceptual problem with using them for any other quantitative variable, in this case ¹⁴C-contents. Other reviewers did not express the same concerns regarding these figures, which leads us to believe that they are generally understandable. We therefore chose to keep the bubble sizes to represent ¹⁴C contents and believe that this is the most effective way to represent our data.

We still made a few editorial changes to the figures to improve the general readability, e.g. juxtapose the bubbles vertically rather than horizontally, and write the specific values to the right of the bubbles (Figure 2&5), add season labeling in each panels and place the legend outside of the plots (Figure 2).

Reviewer #3 (Remarks to the Author):

Review for manuscript number NCOMMS-18-7748888A
Current forest C fixation fuels stream CO₂ emissions

This paper shows that in a boreal, headwater stream, the radiocarbon ages of CO₂ and DOC generally showed little overlap over one year, suggesting that these two carbon pools are primarily sourced from separate biogeochemical pathways that operate over different timescales within the study catchment. Specifically, the ¹⁴C-CO₂ in stream- and groundwater was generally indicative of a contemporary source while ¹⁴C-DOC was more variable and likely originated from an older carbon pool.

This manuscript is an improved version of one I reviewed before. In this revision, the authors have broadened their assignment and interpretation of the modern carbon source feeding streams beyond just root respiration. This change, as well the edits made to the discussion that add interpretation of the flowpaths that deliver CO₂ from catchment soils to streams, have helped to clarify the text and the study findings. I do not have any

substantial hesitations regarding the findings or their interpretation, but offer several minor comments below that may further clarify parts of the manuscript presentation.

We thank the reviewer for his valuable recommendations and the thoroughness of his review.

Specific comments by line:

Line 16: I wonder if “recently fixed” or “contemporary carbon” would be a more fitting word choice throughout the text to replace “current” or “currently fixed.” I leave this up to the authors.

This is an issue that we have long considered throughout the elaboration of this paper. The issues related to the use of the term “recent” and “contemporary” is that they can be interpreted very differently depending on readers’ background, e.g. Modern DOC (>100%modern) could still be regarded as “recent” and “contemporary”. Both the term “recent” and “contemporary” don’t have a specific ending date, making them rather vague and open to interpretation.

We felt that a different term was needed here to refer to even shorter time-scales, hence the choice of “current”. We concede that the term “current” may be over specific in relating to present conditions. However, since the paper makes a direct comparison between the stream/groundwater ¹⁴C-CO₂ and the current atmospheric CO₂, compare lateral CO₂ export with the NEE of that same year (last growing season), we believe that this is a more suitable choice of words and make the terminology more consistent throughout the text.

We made sure that the terminology was consistent throughout the text (e.g. Line183 said recent instead of current). In some instances, we still felt that the interpretation of the results could be broadened a little and therefore opted for the “recent” choice of word in the abstract, Line 16, and when referring to “recent photosynthates” which cannot truly be “current”.

To make this terminology even clearer, we specified on Line 175 (Results) 184 (Discussion) 492 (Methods) that current forest carbon fixation refers to “during the last growing season (2015-2016)”, which was also stated on line 194-195 in earlier versions.

Lines 19-20: Stream CO₂ emissions, specifically, will also be highly dependent on patterns of hydrologic variation (e.g. Liu and Raymond, 2019 - Hydrologic controls on pCO₂ and CO₂ efflux in US streams and rivers), so it is difficult to envision what such responses in stream CO₂ emissions might look like. It may be simpler to state that stream CO₂ fluxes (and/or emissions) will be highly sensitive to patterns of forest C allocation.

Changes made accordingly

Line 41: It is sufficient to say “over longer timescales” here.

Changes made accordingly

Lines 166-167: I am somewhat confused by this statement. Is this comparison between CO₂ export in the upslope and riparian locations made by comparing areal flux rates (as in Figure 5)? If so, comparing these rates in the context of the relative catchment area comprised by riparian soils is not a meaningful comparison compared with export estimates from each zone that are scaled to the whole catchment, for example.

We thank the reviewer for pointing this out. The comparison here was confusing, with C export referring to area specific fluxes and riparian zone coverage estimated for the complete catchment area. We therefore removed the second part of the sentence where we specified the areal coverage of the riparian zone, this information was already available in the method section.

Line 238: Please add more specificity, i.e. rapid C cycling processes within catchment soils.

Changes made accordingly

Line 194: Is “fluxes” the right word here? Consider replacing with “sources.”

Changes made accordingly

Additional change:

- Line 68: we replaced the term re-mineralization of recent photosynthates with “soil respiration” which is a more general and appropriate term.

- A few grammar mistakes were also corrected throughout the text.
- Change reference 23 from ICPP 2013, to 1 Winkler, A. J., Myneni, R. B., Alexandrov, G. A. & Brovkin, V. Earth system models underestimate carbon fixation by plants in the high latitudes. *Nature Communications* **10**, 885, doi:10.1038/s41467-019-08633-z (2019).